# MIDAS: META-WEIGHTED DIFFUSION ALIGNMENT FOR ROBUST MULTIMODAL HUMAN SENSING

## ABSTRACT

Multimodal human sensing systems promise unprecedented accuracy and robustness but are often hindered by a critical real-world challenge: missing modalities, which can occur due to hardware failures, environmental interference (e.g., adverse weather for LiDAR), deployment cost constraints, or communication dropouts. The failure of one or more sensors can severely degrade performance and is exacerbated by two fundamental and intertwined issues: the Representation Gap between heterogeneous sensor data and the Contamination Effect from low-quality modalities. In this paper, we propose MIDAS, a novel framework that tackles both root challenges simultaneously through a synergistic integration of meta-learning and knowledge diffusion. To mitigate the Contamination Effect, MIDAS employs a meta-learning-driven weighting mechanism that dynamically learns to down-weight the influence of noisy, low-contributing modalities. To bridge the Representation Gap, it introduces a diffusion-based knowledge distillation paradigm where an information-rich teacher, formed from all available modalities, refines the features of each student modality. Comprehensive experiments on the large-scale MM-Fi and XRF55 datasets demonstrate that MIDAS achieves state-of-the-art performance, significantly improving robustness in numerous missing-modality scenarios. Our work provides a unified and effective solution for building robust, real-world multimodal human sensing systems.

## 1 INTRODUCTION

Multimodal human sensing, the art of capturing and interpreting human activities and states through various sensors, stands as a cornerstone technology in frontier AI domains such as human-computer interaction, autonomous driving, intelligent healthcare, and the metaverse. The fusion of multiple modalities, for instance, the dense information from depth cameras, the precise 3D structure from LiDAR, and the all-weather perceptual capabilities of wearable devices, offers a path to surmount the limitations of any single sensor, promising systems with superior accuracy and robustness (Baltrušaitis et al., 2018).

However, this vision of an ideal sensing system is often based on a fragile assumption: that all modalities are consistently available. In real-world scenarios, this assumption is frequently violated due to hardware failures, environmental interference (e.g., adverse weather for LiDAR), deployment cost constraints, or communication dropouts (Ma et al., 2021). This issue of missing modalities represents a pervasive challenge that severely degrades the performance of existing fusion models. The problem is further compounded by two fundamental and intertwined challenges:

The first is the Representation Gap. The physical principles of different sensors are profoundly dissimilar, leading to a vast chasm between their data representations (e.g., the grid-like pixels of an image versus the sparse point cloud from LiDAR) (Li et al., 2022). Directly combining these heterogeneous features often results in the loss or misinterpretation of critical information. The second is the Contamination Effect. The signal-to-noise ratio and task-specific contribution of modalities can vary dramatically. When a high-quality modality is fused with a low-quality, high-noise one, the uncertainty from the latter inevitably "contaminates" the former, degrading overall performance. These two challenges are concurrent and mutually exacerbating, forming the primary bottleneck that hinders breakthroughs in robust multimodal learning.

To address these issues, the research community has explored several avenues. Generative approaches, such as VAEs (Kingma & Welling, 2013) or GANs (Goodfellow et al., 2014), attempt to reconstruct missing features but are often plagued by training instability, high computational costs, and a tendency to "hallucinate" inaccurate details. Shared representation learning aims to project all modalities into a common latent space, but often struggles to preserve unique critical information from highly heterogeneous modalities (Liang et al., 2022). Simpler fusion mechanisms like averaging are highly susceptible to the Contamination Effect, while traditional knowledge distillation fails to bridge the immense Representation Gap in multimodal contexts (Xue et al., 2022). In summary, while existing methods have made progress, a unified framework that simultaneously and effectively addresses the Representation Gap and the Contamination Effect remains an open challenge.

To tackle these two challenges simultaneously, we propose a novel paradigm: MIDAS, a framework that integrates meta-learning with knowledge diffusion. To bridge the representation gap, we introduce a knowledge diffusion for a feature refinement strategy. We reframe the distillation problem from mere imitation to alignment. In our novel many-to-one paradigm, all available modalities form a high-quality "consensus teacher". A diffusion model is then trained to purify and align each individual "student" modality by removing its unique biases and noise, aligning it with the consensus teacher. To address the contamination effect, we introduce a meta-learning modality weighting mechanism. This module employs a bi-level optimization process to automatically learn a set of importance weights based on each modality's true contribution to the final task performance. This allows the model to intelligently rely on high-performers and down-weight stragglers during fusion, thereby effectively mitigating the polluting influence of low-quality modalities.

The main contributions of this work are as follows:

- We propose MIDAS, a novel framework that, for the first time, integrates meta-learning and knowledge diffusion to provide a unified solution to the intertwined challenges of the representation gap and contamination effect in multimodal human sensing.

- We introduce a knowledge diffusion-based feature alignment strategy. Our experiments demonstrate that this strategy effectively bridges the representation gap between heterogeneous modalities, which significantly enhances the capability of each single-modality feature encoder.

- We design a meta-learning weighting mechanism that adaptively evaluates the importance of modality based on task performance, effectively mitigating the negative impact of low-quality modalities during fusion.

- We conducted comprehensive experiments on the large-scale multimodal dataset MM-Fi (Yang et al., 2023) and XRF55 (Lan et al., 2025). Our results show that MIDAS achieves state-of-the-art performance in numerous combinations of missing modality.

## 2 RELATED WORK

### 2.1 HUMAN SENSING

Human sensing is a key research area that uses multi-source sensors to recognize human behavior, posture, and physiological states. In vision-based sensing, early foundational works like Open-Pose (Cao et al., 2019) enabled real-time 2D multi-person keypoint detection, while subsequent models like PoseConv3D (Duan et al., 2022) advanced the field by capturing spatio-temporal dynamics for 3D pose estimation directly from video. As a non-intrusive alternative, radio frequency sensing has also seen significant progress. Millimeter-wave signals have been used to reconstruct fine-grained 3D body meshes (Xue et al., 2021) and for gait-based human recognition (Ozturk et al., 2021). Meanwhile, Wi-Fi sensing has been applied to passive activity recognition (Zhang et al., 2022) and human pose estimation (Wang et al., 2019). other researchers have designed custom radar systems for pose estimation (Zhao et al., 2018) and mesh reconstruction (Zhao et al., 2019).

To overcome the weaknesses of single modalities, such as vision's sensitivity to occlusion, the dominant trend is to fuse multimodal data for enhanced robustness and accuracy. This paradigm has led to works combining Wi-Fi, mmWave radar, camera, lidar, and RFID signals (Yang et al., 2023; Wang et al., 2024a)for tasks like human pose estimation and activity recognition, which now defines the state-of-the-art in building reliable, real-world human sensing systems.

## 2.2 Multimodal Learning with Missing Modality

Although multimodal human sensing offers more comprehensive and accurate measurements, in real-world scenarios, some modalities are often unavailable due to sensor instability or hardware deployment costs. Therefore, developing a framework that remains robust under modality-missing conditions has become a critical research direction in this field. Broadly, solutions for this problem follow two strategies. The first is modality generation, which synthesizes raw missing data using methods from simple imputation (Parthasarathy & Sundaram, 2020; Campos et al., 2015; Yang et al., 2024) to generative models like GANs (Gunasekar et al., 2020) and diffusion models (Wang et al., 2023), but they suffer from training instability, high computational costs, and a tendency to "hallucinate" inaccurate details. The second strategy learns collaborative representation relationships in modalities (Wang et al., 2020b; Liu et al., 2021) or directly imputes the representation of missing modalities (Hoffman et al., 2016; He et al., 2022), but It is often plagued by the difficulty of preserving unique and critical information from highly heterogeneous modalities.

In human sensing, these two strategies are usually adapted and complemented by other techniques. For instance, These works (Xue et al., 2022; Kong et al., 2019; Wang et al., 2020a) use knowledge distillation, where transferring knowledge from one powerful multi-modal or many single-modal "teacher" to a "student". This enables the student to either infer the representation of a missing modality or enrich its own features, thus compensating for the incomplete input. However, as mentioned in (Bruce et al., 2021), this can require large pre-trained teachers and may be limited to specific tasks. In contrast, X-Fi (Chen & Yang, 2025) builds a modality-invariant foundational model to learn a compressed representation from available modalities, but it is highly sensitive to the probability of each modality being selected in the training process, leading to unstable results, what's move, it can not bridge the immense Representation Gap in multimodal contexts.

## 3 Method

In this section, we present MIDAS, a novel multimodal human sensing framework designed to address the pervasive challenge of missing modalities (Ma et al., 2021; Wang et al., 2020a). As outlined in Section 1, our approach is architected to simultaneously tackle two fundamental issues: the Contamination Effect, where low-quality modalities degrade overall performance, and the Representation Gap, which arises from the profound dissimilarities between sensor data representations.

Unlike existing methods that rely on fixed or manually tuned modality fusion strategies (von Marcard et al., 2016; Das et al., 2021; Chen & Yang, 2025), MIDAS employs a dynamic, two-pronged strategy. To mitigate the Contamination Effect, we introduce a meta-learning-driven Modality Weighting mechanism (Wang et al., 2024b), which actively and adaptively learns the optimal contribution of each modality to the final task. Concurrently, to bridge the Representation Gap, we design a diffusion-based feature-level knowledge distillation (Ho et al., 2020; Hinton et al., 2015). In this paradigm, a "many-to-one" consensus teacher is leveraged to purify and align the features from each student modality. Through this synergistic design, MIDAS enables the model to maintain robust and accurate performance even when some modalities are unavailable, effectively bridging the performance gap between scenarios with complete and partial multimodal inputs.

### 3.1 Meta-learning-driven Modality Weighting

As shown in Figure 1, our pipeline is designed for robustness in the face of missing modalities. During training, we simulate this by randomly dropping each input modality with a uniform probability. This approach avoids the need for complex, trial-and-error methods to determine optimal dropout rates, as required by prior work such as X-Fi (Chen & Yang, 2025). The available data from each modality is then passed through its respective encoder to yield feature representations. These features are then multiplied by a set of learnable importance weights $\mathbf{w}$ before being fused and passed to a task head for the final human sensing prediction.

To enable the model to dynamically adjust the contribution of each modality based on its relevance and quality, we design a nested optimization strategy based on meta-learning (Ma et al., 2021). This approach decouples the optimization of the meta-parameters $\mathbf{w}$, which control the relative importance of each modality, from the sensing model parameters $\Theta$.

Figure 1: The overall architecture of our proposed MIDAS framework. The model is trained to handle stochastic missing modalities using a two-pronged strategy. (a) Meta-learning-driven Modality Weighting: A process learns importance weights $\mathbf{w}$ to mitigate the Contamination Effect. An inner loop optimizes model parameters, while an outer loop optimizes the weights $\mathbf{w}$. (b) Diffusion-based Knowledge Distillation: To bridge the Representation Gap, a rich teacher feature $f_T$ is constructed from the weighted sum of all available modalities to refine each single-modality student feature.

The optimization process consists of two nested loops:

• Inner loop (training on training set $\mathcal{D}_{\text{train}}$): aims to optimize the main model parameters $\Theta$, including the modality-specific encoders, task heads, and the diffusion-based feature refinement module. During this loop, the modality weights $\mathbf{w}$ are treated as fixed hyperparameters. The inner loop ensures that the model learns a robust mapping from any subset of active modalities to task-relevant predictions.

• Outer loop (training on a disjoint validation set $\mathcal{D}_{\text{val}}$): aims to optimize the modality weight meta-parameters $\mathbf{w}$ by evaluating the performance of the inner-loop-trained model on an independent validation set. The model parameters $\Theta$ are treated as fixed while updating $\mathbf{w}$. This enables the model to adaptively adjust the relative importance of each modality to maximize generalization.

The objective of the inner loop is to optimize the sensing model parameters $\Theta$. This is achieved by minimizing a composite loss function that combines a standard downstream task loss with with our proposed feature alignment loss (detailed in Section 3.2).

$$\mathcal{L}_{\text{inner}} = \mathcal{L}_{\text{task}}(\Theta) + \lambda \mathcal{L}_{\text{DiffKD}}(\Theta), \tag{1}$$

where $\lambda$ balances the contribution of the feature alignment term. This ensures that the model not only learns to excel at the downstream task but also produces robust and well-aligned feature representations across different modalities, which is crucial when some modalities are missing.

After inner-loop updates, $\Theta$ fixed as $\Theta^*$, the outer loop evaluates the model's performance on $\mathcal{D}_{\text{val}}$ and updates the modality weights $\mathbf{w}$ according to the outer-loop loss:

$$\mathcal{L}_{\text{outer}} = \mathcal{L}_{\text{task}}(\Theta^*(\mathbf{w})). \tag{2}$$

The gradient $\nabla_{\mathbf{w}} \mathcal{L}_{\text{outer}}$ is then computed to update the modality weights. Formally, this nested optimization problem can be expressed as:

$$\mathbf{w}^* = \arg\min_{\mathbf{w}} \sum_{(\mathcal{M}_{\text{val}}, y) \in \mathcal{D}_{\text{val}}} \mathcal{L}_{\text{task}}\left( h^*\left( \sum_{i \in \mathcal{M}_{\text{val}}} w_i f_i'^* \right), y \right) \tag{3}$$

$$\text{s.t.} \quad \Theta^* = \arg\min_{\Theta} \sum_{(\mathcal{M}_{\text{train}}, y) \in \mathcal{D}_{\text{train}}} \left[ \mathcal{L}_{\text{task}}\left( h\left( \sum_{i \in \mathcal{M}_{\text{train}}} w_i f_i \right), y \right) + \lambda \mathcal{L}_{\text{DiffKD}}(\Theta) \right] \tag{4}$$

where $h$ and $h^*$ represent the task-specific before and after the inner-loop update, respectively. $f_i$ and $f_i'^*$ are the original and distilled features from modality $i$. Finally, $\mathcal{M}_{\text{train}}$ and $\mathcal{M}_{\text{val}}$ are the subsets of available modalities for the training and validation steps.

To ensure stability and interpretability, the weights $\mathbf{w}$ are normalized via a Softmax function after each outer-loop update:

$$\mathbf{w}_i \leftarrow \frac{\exp(\mathbf{w}_i)}{\sum_{j=1}^{N} \exp(\mathbf{w}_j)}, \quad i = 1, \ldots, N. \tag{5}$$

This meta-learning framework allows the model to jointly learn task-specific representations and modality importance in a principled, adaptive manner, reducing the contaminating effects of low-quality modalities on the fused representation.

## 3.2 DIFFUSION-BASED KNOWLEDGE DISTILLATION FOR FEATURE ALIGNMENT

To bridge the Representation Gap, we design a Diffusion-based Feature Alignment mechanism that integrates knowledge distillation with diffusion models (Ho et al., 2020; Hinton et al., 2015; Huang et al., 2023). As shown in Figure 1, this is a "many-to-one" paradigm, where the "teacher" feature, derived from a consensus of all available modalities, is used to guide the "student" feature alignment from each modality. This process effectively purifies and aligns the features from each student, encouraging them to approximate the comprehensive representation of the consensus teacher. Next, we will introduce the important technical details.

**Teacher Feature and Student Feature.** The teacher feature, $f_T$, is constructed to represent the optimal fusion of all available modalities that serves as the distillation target. It is formally defined as the weighted sum of the features from all available modalities in the set $\mathcal{M}_{\text{all}}$:

$$f_T = \sum_{i \in \mathcal{M}_{\text{all}}} \mathbf{w}_i f_i, \tag{6}$$

where $f_i$ is the feature of modality $i$ after passing through its specific encoder and a projection layer, and $\mathbf{w}_i$ is its corresponding meta-learned importance weight. This construction ensures that $f_T$ is a comprehensive and intelligently fused representation. In contrast, the student feature, $f_S$, is defined as the encoded and projected feature of any single available modality. The goal of our alignment process is to transform the information-deprived $f_S$ from a single modality into a feature that structurally aligns with the rich teacher feature $f_T$.

**Diffusion-based Feature Refinement.** To align the student feature $f_S$ with the teacher feature $f_T$, we employ a diffusion-based framework, conceptualizing $f_S$ as a "noisy" or incomplete version of $f_T$. The core of this process is a noise prediction network, $\Phi_\phi$, trained exclusively on the 'clean' teacher features $f_T$ to learn their underlying data distribution. The training objective is the standard diffusion loss, $\mathcal{L}_{\text{Diff}}$:

$$\mathcal{L}_{\text{Diff}} = \mathbb{E}_{t, f_T, \epsilon_t} \left[ ||\epsilon_t - \Phi_\phi(z_t, t)||_2^2 \right], \quad \text{where } z_t = \sqrt{\bar{\alpha}_t} f_T + \sqrt{1 - \bar{\alpha}_t} \epsilon_t. \tag{7}$$

Here, $t$ is the diffusion timestep, $\epsilon_t \sim \mathcal{N}(0, \mathbf{I})$ is the sampled Gaussian noise, and $\bar{\alpha}_t$ is a parameter derived from the predefined noise schedule.

A key challenge, however, is the inherent ambiguity in the student feature "noise level" Since $f_S$ is an incomplete representation, we cannot assume it corresponds to a specific timestep $t$ in the diffusion process. To address this, we introduce a lightweight **Noise Adapter**. This module's role is to resolve the ambiguity by mapping $f_S$ to a well-defined starting point for the reverse (denoising) process, typically at the maximum timestep $T$. Architecturally, the adapter is a small convolutional network that predicts a fusion coefficient, $\gamma$. This coefficient is then used to optimally blend the student feature with pure Gaussian noise $\epsilon_T$:

$$z_{TS} = \gamma f_S + (1 - \gamma)\epsilon_T, \quad \text{where } \epsilon_T \sim \mathcal{N}(0, \mathbf{I}). \tag{8}$$

This resulting feature, $z_{TS}$, serves as the high-quality starting point for the reverse denoising process. This generative process is executed using the Denoising Diffusion Implicit Models (DDIM) sampling strategy (Song et al., 2021), which accelerates inference by taking larger, deterministic steps, thereby efficiently yielding the aligned and refined student feature, $f_S'$.

**Overall Loss.** The total feature alignment loss is the sum of the diffusion loss and knowledge distillation loss:

$$\mathcal{L}_{\text{DiffKD}} = \mathcal{L}_{\text{Diff}} + \mathcal{L}_{KD}. \tag{9}$$

This design ensures that even when only partial modalities are available, the student feature is refined to approximate a complete multimodal representation, improving robustness and modality-invariance.

### 3.3 STOCHASTIC MISSING MODALITY TRAINING

To simulate realistic scenarios with missing modalities, we randomly sample a subset of active modalities, $\mathcal{M}_{\text{active}} \subseteq \mathcal{M}_{\text{all}}$, at each training iteration for both feature alignment and task supervision. The modality-specific encoders are pre-trained and fine-tuned to ensure that their learned representations are stable. This strategy allows the meta-learning and diffusion processes to focus on the higher-level tasks of feature fusion and alignment rather than low-level feature extraction.

At inference time, the model makes predictions by combining the available modalities, denoted by the set $\mathcal{M}_{\text{infer}}$, using the optimal meta-learned weights $\mathbf{w}^*$. Specifically, the final prediction $\hat{y}$ is generated by first computing a weighted average of the refined features from each available modality. This aggregated feature is then passed through the optimized task-specific head $h^*$ to produce the final output. This process is formalized as:

$$\hat{y} = h^* \left( \sum_{i \in \mathcal{M}_{\text{infer}}} \mathbf{w}_i^* f_i'^* \right) \tag{10}$$

It is worth noting that the computationally intensive diffusion model training and meta-weight optimization are performed only once during the training phase. At inference time, the model simply applies these pre-learned components. This ensures that the framework can generalize to arbitrary combinations of missing modalities while maintaining low computational overhead (reported in the supplementary materials) and producing robust sensing results (reported in Section 4).

## 4 EVALUATION

### 4.1 DATASETS

**MM-Fi** (Yang et al., 2023) is a large-scale, multimodal dataset designed for non-intrusive human pose estimation. It provides more than 320,000 synchronized frames collected from 40 human subjects performing 27 distinct actions. The dataset includes three publicly available sensing modalities: depth image, LiDAR point cloud, and Wi-Fi channel state information (CSI) data. This rich combination of high and low-resolution sensors makes MM-Fi an ideal benchmark for evaluating the robustness of multimodal human sensing frameworks, especially under challenging missing modality conditions. We used the official data splits and evaluation protocols for our experiments.

**XRF55** (Wang et al., 2024a) is another large-scale, multi-modal radio frequency (RF) dataset designed for human indoor action recognition. It contains 42,900 RF samples, totaling over 59 hours of data, collected from 39 human subjects performing 55 distinct actions, covering categories such as fitness activities and human-computer interactions. The dataset's key feature is its use of diverse, commercial off-the-shelf RF sensing modalities, including one mmWave radar (60-64GHz) providing Range-Doppler & Range-Angle Heatmaps (R), 9 Wi-Fi links capturing Wi-Fi CSI at 5.64GHz (W), and 23 RFID tags providing phase series data at 922.38 MHz. in our experiments, we follow the original data split setting for XRF55 as outlined in its paper.

### 4.2 IMPLEMENTATION DETAILS

**On MM-Fi dataset.** For the human pose estimation (HPE) task on the MM-Fi dataset, the modality encoders for the Depth, LiDAR, and Wi-Fi modalities are based on pre-trained and frozen models in X-Fi (Chen & Yang, 2025). We standardize the feature representation of each modality to a dimension of 512 ($d_f = 512$). The noise prediction network within the knowledge diffusion module is implemented as a lightweight residual network, composed of two consecutive bottleneck blocks

followed by a final 1D convolutional layer; the feature refinement itself is performed using a DDIM sampling strategy with 5 inference steps. Our MIDAS framework is optimized using two separate Adam optimizers: one for the main model parameters with a learning rate of $5 \times 10^{-4}$, and one for the meta-learning weights with a learning rate of $1 \times 10^{-2}$. The balancing coefficient $\lambda$ for the knowledge distillation loss is set to 0.1. All experiments are conducted on an NVIDIA RTX 3090 GPU with a batch size of 16. We use the Mean Per Joint Position Error (MPJPE) and Procrustes-Aligned MPJPE (PA-MPJPE) as evaluation metrics, where lower values indicate better performance.

**On XRF55 dataset.** For the human action recognition (HAR) task on the XRF55 dataset, the modality encoders for all RF modalities (mmWave Radar, Wi-Fi CSI, and RFID) are based on a pre-trained and frozen ResNet-18 architecture. We standardize the feature representation of each modality to 32 features ($n_f = 32$), each with a dimension of 512 ($d_f = 512$). The noise prediction network and refinement strategy are identical to those used for the MM-Fi dataset. Similarly, our MIDAS framework is optimized using the Adam optimizer with a learning rate of $2 \times 10^{-4}$ and $\lambda = 0.1$. All experiments are conducted on an NVIDIA RTX 3090 GPU with a batch size of 32. We use classification accuracy as the evaluation metric, where higher values indicate better performance.

### 4.3 HUMAN POSE ESTIMATION RESULTS ON THE MM-FI DATASET

We compare MIDAS against three key methods: a feature-fusion baseline where modality-specific features are concatenated and processed by an MLP fusion head (Das et al., 2021) (Base.1), a decision-level baseline that averages the final prediction outputs from each modality (Yang et al., 2023) (Base.2), and the state-of-the-art modality-invariant model, X-Fi (Chen & Yang, 2025).

Table 1: MPJPE (mm) and PA-MPJPE (mm) on MM-Fi (Yang et al., 2023) dataset.

| Modality | Base.1 | Base.2 | X-Fi | Ours | Imp↑ | Base.1 | Base.2 | X-Fi | Ours | Imp↑ |
|---|---|---|---|---|---|---|---|---|---|---|
| | | | MPJPE ↓ | | | | | PA-MPJPE ↓ | | |
| D | 102.4 | 102.4 | 96.40 | **84.81** | +12.0% | 52.7 | 52.7 | **48.76** | 50.72 | -4.0% |
| L | 161.5 | 161.5 | 130.06 | **68.30** | +47.5% | 103.5 | 103.5 | 89.83 | **44.44** | +50.5% |
| W | 227.1 | 227.1 | 210.12 | **182.18** | +13.3% | 108.0 | 108.0 | **103.91** | 114.63 | -10.3% |
| D+L | 111.7 | 108.0 | 89.41 | **64.68** | +27.7% | 68.8 | 55.8 | 47.97 | **42.63** | +11.1% |
| D+W | 141.7 | 155.5 | **95.27** | 95.96 | -0.7% | 71.4 | 81.5 | **47.71** | 59.77 | -25.3% |
| L+W | 167.1 | 206.2 | 111.15 | **74.74** | +32.8% | 100.7 | 109.3 | 75.11 | **49.49** | +34.1% |
| D+L+W | 130.7 | 154.6 | 87.59 | **68.86** | +21.4% | 78.1 | 84.1 | 47.52 | **45.47** | +4.3% |
| Best Count | 0 | 0 | 1 | **6** | - | 0 | 0 | 3 | **4** | - |

**Finding 1: Knowledge Diffusion is Powerful for Bridging the Representation Gap.**

As shown in Table 1, our method significantly boosts the performance of each modality in isolation when compared to both the baseline and X-Fi. For instance, the performance of the Depth, LiDAR, and Wi-Fi CSI modalities individually improved by (Imp) 12.0%, 47.5%, and 12.3%, respectively, over X-Fi. This substantial improvement is a direct result of our core mechanism: MIDAS enriches the features of each modality by aligning them with an information-rich teacher feature derived from a consensus of all modalities. In contrast, X-Fi primarily focuses on the co-adaptation of features for fusion, which does not yield a comparable improvement in the intrinsic quality of single-modality representations. By compelling each single-modality student to be refined by a higher-quality teacher, our framework dramatically enhances the capability of each feature extractor.

**Finding 2: Meta-Learning Weighting Mitigates the Contamination Effect.**

As shown in Table 1, the Wi-Fi modality, with a standalone MPJPE of 182.18, performs significantly worse than Depth (84.81) and LiDAR (68.30), establishing it as a low-quality modality for this task. The introduction of such a modality risks contaminating the fused representation and degrading overall performance. Our results demonstrate that the meta-learning weighting mechanism effectively mitigates this risk. For instance, when fusing LiDAR with Wi-Fi (L+W), our method achieves an MPJPE of 74.74, only a minor degradation from LiDAR's standalone performance of 68.30. This indicates that MIDAS successfully identified Wi-Fi as a low-contributor and down-weighted its influence, thus preventing significant contamination.

Interestingly, X-Fi's performance for L+W (111.15) is better than its standalone LiDAR performance (130.06). However, this is a consequence of the unfavorable performance trade-off inherent in its design. X-Fi sacrifices the maximum performance of its base single-modality models (as evidenced by its LiDAR MPJPE of 130.06 vs. our 68.30) in exchange for marginal fusion gains. Our results suggest this is an inefficient trade-off, as our method's superior single-modality performance provides a much stronger foundation, leading to a more favorable and robust overall performance.

**Discussion on MPJPE vs. PA-MPJPE.**

An interesting trade-off is observed between the absolute error (MPJPE) and structure-aligned error (PA-MPJPE). Our method's superior MPJPE performance stems from the knowledge diffusion process effectively extracting and enhancing global positioning information from noisy signals like CSI, allowing the model to better determine where a person is. However, these same signals lack the local details required for precise limb placement. Consequently, after the Procrustes Alignment in PA-MPJPE nullifies the global advantage, the evaluation focuses on the skeleton's internal structure. Lacking fine-grained local information, our model may converge on a structurally plausible but incorrect "average pose", causing a slight increase in the PA-MPJPE. Future hybrid approaches that integrate biomechanical constraints or pose priors could potentially mitigate this issue.

### 4.4 HUMAN ACTION RECOGNTION RESULTS ON THE XRF55 DATASET

We compare MIDAS with two methods: a mutual learning method implemented XRF55 (Wang et al., 2024a) (Base.) and the state-of-the-art X-Fi (Chen & Yang, 2025).

**Echoing Finding 1: Knowledge Diffusion Bridges the Representation Gap.**

Table 2: Accuracy (%) on the XRF55 dataset.

| Modality | Base. | X-Fi | Ours | Imp ↑ |
|---|---|---|---|---|
| R | 82.1 | 83.9 | **90.03** | +6.13 |
| W | 77.8 | 55.7 | **82.34** | +26.64 |
| RF | 42.2 | 42.5 | **55.04** | +12.54 |
| R+W | 86.8 | 88.2 | **95.23** | +7.03 |
| R+RF | 71.4 | 86.5 | **92.15** | +5.65 |
| W+RF | 55.6 | 58.1 | **83.14** | +25.04 |
| R+W+RF | 70.6 | 89.8 | **95.87** | +6.07 |
| Best Count | 0 | 0 | **7** | - |

As shown in Table 2, MIDAS demonstrates a significant performance increase for single modalities compared to X-Fi. Specifically, the mmWave Radar (R), Wi-Fi (W), and RFID (RF) modalities achieve accuracy improvements of +6.13%, +26.64%, and +12.54%, respectively. This significant uptick in performance is a direct result of our knowledge diffusion mechanism, which echoes the core principle of MM-Fi. Our system trains the feature extractor of each modality to align its representation with a comprehensive teacher feature, which is synthesized from the consensus of all modalities. This process effectively allows less descriptive modalities, such as Wi-Fi, to learn from the more robust and detailed patterns captured by others.

**Finding 2: Meta-Learning Mitigates Contamination and Can Yield Positive Fusion Gains.**
In the MM-Fi results, we observed that a low-quality modality could contaminate a high-quality one, leading to fusion performance that was worse than the high-quality modality alone. However, our findings on XRF55 are more optimistic and show this is not always the case. As shown in Table 2, RFID is the weakest modality with a standalone accuracy of only 55.04%. Yet, when fused with Radar, it improves the accuracy from 90.03% to 92.15%. Similarly, fusing it with Wi-Fi improves accuracy from 82.34% to 83.14%. Furthermore, when all modalities are present, our performance reaches 95.87%, significantly outperforming the baseline (70.6%) and X-Fi (89.8%). This demonstrates that under certain conditions, our framework effectively resists the Contamination Effect and leverages weak modalities to achieve performance gains.

We hypothesize this occurs because the modalities in XRF55 (mmWave, Wi-Fi, RFID) are all RF-based signals. The representation gap between them is smaller than the gap between the heterogeneous modalities in MM-Fi (Wi-Fi, Depth, LiDAR), allowing MIDAS to learn more effective alignment and fusion strategies. An open question remains regarding the threshold at which a modality becomes so poor that it actively contaminates others. While RFID is the weakest performer here, it may not be weak enough to cause significant harm, unlike the CSI data in MM-Fi. This aligns with findings in NLP where "dirty" data can degrade performance, but it remains a less-explored and intriguing area for future research in the human sensing domain.

### 4.5 ABLATION STUDY

To validate the effectiveness of the core components within our proposed MIDAS framework, we conducted a series of rigorous ablation studies on both the MM-Fi and XRF55 datasets. We systematically removed the Diffusion-based Alignment module and the meta-learning weighting module to independently evaluate their respective contributions. The results are presented in Table 3 and Table 4, respectively.

Table 3: Ablation study of our MIDAS model on the MM-Fi HPE task, (MPJPE (mm)).

| Modality | Full | w/o Diff. | w/o Meta. |
|---|---|---|---|
| D | **84.81** | 89.66 | 157.98 |
| L | **68.30** | 76.27 | 183.04 |
| W | **182.18** | 187.92 | 236.99 |
| D+L | **64.68** | 78.12 | 148.65 |
| D+W | 95.96 | **90.14** | 179.77 |
| L+W | 74.74 | **74.23** | 189.24 |
| D+L+W | **68.86** | 76.79 | 160.34 |
| Best count | **5** | 2 | 0 |

Table 4: Ablation study of our MIDAS model on the XRF55 HAR task, (Accuracy%).

| Modality | Full | w/o Diff. | w/o Meta. |
|---|---|---|---|
| R | **90.03** | 89.17 | 89.04 |
| W | **82.34** | 81.07 | 79.56 |
| RF | **55.04** | 54.49 | 54.79 |
| R+W | 95.23 | **95.40** | 93.61 |
| R+RF | 92.15 | 91.60 | **92.87** |
| W+RF | 83.14 | **84.35** | 78.35 |
| R+W+RF | **95.87** | 95.59 | 95.07 |
| Best count | **4** | 2 | 1 |

The results in Table 3 and Table 4 confirm that the full MIDAS framework consistently achieves the best performance, securing the top result in 5 out of 7 modality combinations on the MM-Fi human pose estimation task and 4 out of 7 on the XRF55 human action recognition task. This demonstrates that the two modules are not merely independent components but are synergistically integrated. They work together effectively to enhance performance under missing modality conditions, forming a cohesive and powerful framework.

When the Diffusion-based Alignment module is removed ("w/o Diff."), We observe a notable performance drop, particularly in single-modality scenarios (e.g., LiDAR MPJPE increases from 68.30 to 76.27). The effect is even more pronounced when the meta-learning weighting module is removed ("w/o Meta."). In this case, performance degrades catastrophically across nearly all fusion scenarios (e.g., D+L MPJPE skyrockets from 64.68 to 148.65), providing clear evidence of its success in mitigating the Contamination Effect by managing the influence of disparate modalities.

A more nuanced analysis of the ablation studies reveals an interesting edge case. While the Diffusion-based Alignment module is overwhelmingly beneficial, we observed a few specific scenarios where its removal paradoxically improved performance. As shown in Table 3 (L+W) and Table 4 (W+RF), this phenomenon occurs when fusing two modalities where at least one is of significantly low quality (Wi-Fi or RFID). We hypothesize that this is due to the generative nature of the diffusion process. While the model's primary function is to align features and bridge the representation gap, the reverse sampling process can still introduce minor generative artifacts or amplify inherent noise, especially when the conditioning student feature is extremely noisy or information-poor.

## 5 CONCLUSION

In this work, we introduced MIDAS, a novel framework designed to address two fundamental and intertwined challenges in multimodal human sensing: the Representation Gap and the Contamination Effect. Our approach pioneers a synergistic combination of a meta-learning weighting mechanism and a diffusion-based knowledge distillation strategy. The experimental results on the MM-Fi and XRF55 datasets robustly demonstrate the efficacy of our two-pronged approach.

Our findings indicate that tackling these core challenges with dedicated, complementary mechanisms is a highly effective strategy for building robust multimodal human sensing systems. The success of MIDAS not only sets a new state-of-the-art beyond X-Fi (Chen & Yang, 2025) for human sensing with missing modalities but also offers a principled methodology for fusing heterogeneous sensors. Future work could explore the theoretical boundaries of the contamination effect, investigating the precise threshold at which a weak modality transitions from being a beneficial contributor to a detrimental polluter.

## ETHICS STATEMENT

This research is committed to responsible technological innovation. We utilize the public datasets MM-Fi and XRF55, strictly adhering to their usage protocols and protecting data privacy. We recognize that human sensing technologies, particularly those using Wi-Fi or mmWave radar, carry a risk of being used for unauthorized surveillance. However, our research aims to enhance robustness for benign applications such as intelligent healthcare and autonomous driving. Concurrently, we note that the datasets used may have inherent demographic biases, which can be inherited by the model. We encourage future work to address the resulting fairness issues to ensure the technology is inclusive.

## REPRODUCIBILITY STATEMENT

Code and scripts are provided in the supplementary materials to replicate the empirical results.

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

# A  APPENDIX

## (1) Our method requires to train less parameters than X-Fi.

Table 5: Comparison of parameter counts for the MIDAS and X-Fi blocks, excluding the feature encoders.

| Model Structure | | Number of Parameters |
|---|---|---|
| X-Fi | (3 modalities) | 11,593,779 |
| | (2 modalities) | 8,438,835 |
| | (1 modality) | 5,283,891 |
| MIDAS (Ours) | | **4,954,434** |

A noteworthy advantage of our proposed MIDAS framework lies in its parameter efficiency. As shown in Table 5, unlike X-Fi, whose parameter count grows substantially with the number of modalities (e.g., from 5.28M for a single modality to 11.59M for three modalities), MIDAS maintains a fixed and compact parameterization of only 4.95M. This modality-invariant design not only reduces memory and computational overhead but also highlights the scalability of our approach when extending to diverse multimodal settings.

## (2) Qualitative Results for Human Pose Estimation (HPE) 3D Visualization of Pose Estimation Results

Table 6: Qualitative results with different modality combinations.

| Modality | Sample 1 | Sample 2 | Sample 3 | Sample 4 |
|---|---|---|---|---|
| **Ground Truth** | | | | |
| D | | | | |
| L | | | | |
| W | | | | |

Table 6: Qualitative results (continued).

| Modality | Sample 1 | Sample 2 | Sample 3 | Sample 4 |
|----------|----------|----------|----------|----------|
| D + L | | | | |
| D + W | | | | |
| L + W | | | | |
| D + L + W | | | | |

As shown in Table. 6, we present the human pose estimation results for the action "raising both hands" under different modality configurations. Each set of results consists of four consecutive frames for comparison with the ground truth. From the single-modality results, it can be observed that the L modality exhibits inaccuracies in capturing lower-limb movements, while the W modality shows deficiencies in modeling upper-limb movements. Moreover, even in the fused D+W configuration, the estimation of upper-limb movements remains suboptimal. In contrast, integrating multiple modalities effectively mitigates the limitations of individual modalities, leading to predictions that are more closely aligned with the ground truth.

USE OF LARGE LANGUAGE MODELS

In the preparation of this manuscript, we employed Large Language Models (LLMs) as a writing assistant. The primary application was to enhance the overall quality of the text by improving sentence fluency, ensuring grammatical correctness, and eliminating typographical errors. Furthermore, we leveraged LLMs to help formulate a more in-depth analysis of our findings, which assisted in making the interpretation of our results more insightful and consistent with the observed data.

