# OpenReview forum: "Midas: Meta-weighted Diffusion Alignment for Robust Multimodal Human Sensing"
_ICLR.cc/2026/Conference — ICLR 2026 Conference Withdrawn Submission_

### Official Review · Reviewer_rwyZ · 2025-10-30

**Soundness:** 1
**Presentation:** 1
**Contribution:** 1
**Rating:** 2
**Confidence:** 3

**Summary:**

This paper proposes the MIDAS framework to address two challenges in human sensing scenarios: the representation gap between different modality data and the varying quality across modalities. To tackle these issues, the paper introduces two solutions. First, meta-learning employs a set of learnable weights and modality dropout during training to simultaneously learn modality differences and handle missing modality scenarios. Second, diffusion-based knowledge distillation incorporates a diffusion step into knowledge distillation to better bridge the representation gap. Experiments are conducted on two datasets comparing the proposed method against three baselines under various missing modality scenarios, showing that the proposed method outperforms baselines in most cases.

**Strengths:**

I am uncertain about the strengths of this paper.

Based on ICLR's evaluation dimensions (originality, quality, clarity, and significance), I assess this work as follows:

1. Originality: The research problem, experimental setup, and a substantial portion of experimental results are copied from X-Fi (Chen & Yang, 2025).

2. Quality: The experimental methodology is questionable, experiments are not convincing, and experiments are hard to reproduce.

3. Clarity: The paper lacks clarity. There is no formal problem formulation, no algorithmic pseudocode, and insufficient specification of experimental settings, among others.

4. Significance: The contribution is insignificant. Given the unconvincing experiments, I do not believe this work will advance related communities.

**Weaknesses:**

1. The most critical and unacceptable issue with this paper is that the experiments are unconvincing:
    - A lot of experimental results are directly copied from the X-Fi (Chen & Yang, 2025) paper, particularly all "Base.1" and "Base.2" results. I personally find this unacceptable as different experimental setups may lead to unfair comparisons. While I am not able to examine the code from both papers in exhaustive detail and many settings are not clearly specified in the papers, I notice that the optimizer, learning rate, and training epochs appear to be different.
    - Comparing Table 1 in X-Fi with Table 1 in this paper, the I and R modalities are missing. Why are these modality combinations not compared? If the proposed method is trained on only the D, L, and W modalities while the baseline methods are trained on all 5 modalities, this is clearly an unfair comparison.

2. The proposed method is merely wrapped in elaborate language without offering anything new:
    - Meta-learning: This is essentially learning weighted averaging weights for modality features.
    - Stochastic missing modalities training: This is simply modality dropout.
    - Diffusion-based knowledge distillation: This just adds a diffusion model before feature alignment.

    The first two are existing methods, and the last one's contribution is difficult to understand. Moreover, the ablation study demonstrates that the last component has little impact.

**Questions:**

1. I believe Table 5's direct comparison between X-Fi's parameter count and the proposed method's model parameters is unreasonable, as they have completely different structures. The former is transformer while the latter is diffusion. Therefore, parameter count does not equal computational efficiency. It would be better to measure runtime on the same device.

2. The paper claims "The balancing coefficient $\lambda$ for the knowledge distillation loss is set to 0.1." What is the rationale? Why does Eq. (9) not require a hyperparameter to balance the two losses?

3. I suggest that this paper provide pseudocode describing the entire training algorithm, especially given that Figure 1 shows two inner loops and one outer loop, whose sequential relationship is sometimes very difficult to understand.

4. The proposed method does not appear to be specific to human sensing scenarios, and it merely uses human sensing datasets. I am uncertain why the paper heavily emphasizes human sensing, and what unique challenges of human sensing are being addressed here.

5. More clarification and experiments are necessary, such as different diffusion model architectures, hyperparameter settings, runtime analysis, etc. In addition, all current experiments assume that modality availability is known at the inference stage, but this may not hold in the real world. As the authors mention in the abstract and introduction regarding hardware failures and environmental interference, users may not be supposed to know that certain modalities are unavailable. How does the model perform when these modality data contain noise, occlusions, and other corruptions?

---

### Official Review · Reviewer_hcgj · 2025-10-31

**Soundness:** 2
**Presentation:** 3
**Contribution:** 2
**Rating:** 4
**Confidence:** 2

**Summary:**

This paper proposes MIDAS, a novel framework that integrates meta-learning and knowledge diffusion to provide a unified solution to the intertwined challenges of the representation gap and contamination effect in multimodal human sensing. Experiments are conducted on two large-scale multimodal dataset MM-Fi and XRF55, demonstrating that MIDAS achieves SOTA performance, and improves robustness in numerous missing-modality scenarios.

**Strengths:**

1. The idea of using diffusion models to denoise features and align them across different modalities is innovative.
2. Experiments demonstrate the effectiveness of the proposed method.

**Weaknesses:**

1. The paper primarily focuses on human pose estimation and action recognition tasks, with limited evaluation of the method's performance on other general multimodal missing modality tasks. For example, tasks like multimodal sentiment analysis, which involves common modalities such as vision, language, and audio, are not considered.
2. The paper's efficiency analysis for the diffusion model is insufficient. While the paper mentions using 5-step DDIM sampling for inference and parameter count, this is not enough to assess its full computational cost. Key training and inference details are missing. For example,  the total number of denoising steps during the diffusion training process is not reported, and the paper lacks a quantitative analysis of the diffusion model's inference efficiency.
3. The paper mentions that modality weights are adjusted through a meta-learning process on a validation set. However, in real-world applications, especially in open set scenarios, a validation set may not be available.
4. The majority of the references in the paper are from before 2023.

**Questions:**

1. I am curious about how the proposed method performs on more general multimodal tasks, such as multimodal sentiment analysis, which involves a broader set of common modalities like vision, language, and audio.
2. Some existing works use diffusion models to recover missing modality features [1][2]. Could the authors clarify what advantages MIDAS has over these approaches?
3. The paper mentions that diffusion models are used to denoise features and align modality representations. However, there is a lack of quantitative analysis and comparative experiments on the denoising effectiveness.

[1] Kebaili A, Lapuyade-Lahorgue J, Vera P, et al. AMM-Diff: Adaptive Multi-Modality Diffusion Network for Missing Modality Imputation[C]//2025 IEEE 22nd International Symposium on Biomedical Imaging (ISBI). IEEE, 2025: 1-4.
[2] Dai R, Li C, Yan Y, et al. Unbiased Missing-modality Multimodal Learning[C]//Proceedings of the IEEE/CVF International Conference on Computer Vision. 2025: 24507-24517.

---

### Official Review · Reviewer_g746 · 2025-10-31

**Soundness:** 3
**Presentation:** 3
**Contribution:** 2
**Rating:** 6
**Confidence:** 3

**Summary:**

This paper addresses the missing modality problem in multimodal human sensing systems, where one or more sensor inputs, such as LiDAR, depth, or Wi-Fi, may be unavailable or corrupted due to hardware failures or environmental noise. The authors identify two intertwined challenges: the Representation Gap between heterogeneous sensing modalities, and the Contamination Effect, where low-quality modalities degrade the fused representation.

To tackle these challenges, the paper proposes MIDAS, a unified framework that integrates two components: a meta-learning driven modality weighting mechanism that dynamically learns the importance of each modality to mitigate the contamination effect, and a diffusion-based knowledge distillation strategy that refines each single-modality feature by aligning it with a multi-modal teacher representation, thereby bridging the representation gap.

Experiments on two large-scale multimodal human sensing datasets, MM-Fi for 3D human pose estimation and XRF55 for human action recognition, show that MIDAS achieves state-of-the-art performance, outperforming prior baselines such as X-Fi.

**Strengths:**

The paper addresses an important and practical problem in multimodal human sensing—robust learning under missing or degraded modalities. The motivation is clear and well-grounded in real-world sensing scenarios, and the paper provides a well-structured formulation that directly targets two key challenges, the representation gap and the contamination effect, with corresponding and well-matched solutions.

The proposed framework is conceptually sound and clearly presented. The overall writing is smooth and logically organized, making the technical ideas easy to follow. The introduction and methodology sections provide a coherent motivation-to-solution flow.

The introduction of the Noise Adapter is particularly interesting. It offers an elegant way to estimate how noisy each student modality is before applying diffusion refinement, effectively addressing the uncertainty of the initial noise level without additional supervision.

The stochastic modality dropout strategy is a fair and practical improvement over prior methods such as X-Fi, since it avoids hand-tuning a dropout rate hyperparameter while still ensuring diverse missing-modality configurations during training.

Experimental results are promising and consistently support the claims.

**Weaknesses:**

While the paper presents an interesting motivation for replacing generative approaches such as VAEs and GANs with a diffusion-based knowledge distillation strategy, it is not entirely clear whether the proposed method fully overcomes the same limitations. Diffusion models can also suffer from high computational costs and long training times, and the paper would benefit from a clearer discussion or quantitative comparison of training efficiency and stability relative to these baselines.

Another limitation is that the paper identifies the preservation of unique and modality-specific critical information as an open challenge, but this aspect is not directly evaluated or clearly demonstrated in the paper. It would strengthen the work to include analyses or studies that directly measure information preservation and show how well the proposed diffusion alignment maintains modality-specific cues, rather than only reporting performance on downstream tasks.

**Questions:**

1. The paper mentions that both the meta-learning based modality weighting and the diffusion-based knowledge distillation modules are computationally involved. Could you provide more details about the training cost and efficiency? For instance, how much additional computation or time is required compared to X-Fi or other baselines?

2. Since the diffusion alignment encourages each student modality to move closer to the fused teacher representation, could this process risk erasing the modality-specific characteristics? It would be helpful to clarify how the method preserves unique and critical modality information.

3. In the results discussion, Table 1 suggests that X-Fi achieves decreased MPJPE when adding the weak Wi-Fi modality to LiDAR (L → L+W), whereas MIDAS shows the opposite trend. Could the authors clarify why MIDAS appears to be more sensitive to the addition of a weak or noisy modality, and whether this reflects limitations of the meta-learning weighting or the diffusion alignment mechanism?

4. The diffusion refinement is performed with only five DDIM sampling steps. How was this number chosen, and is there any empirical justification that five steps are sufficient? A sensitivity analysis on the number of sampling steps would make this design choice more convincing.

5. Have the authors considered adding generative approaches such as GANs or VAEs as additional baselines for comparison? Including such models could highlight the efficiency and stability advantages of the proposed diffusion-based method when handling missing modalities.

---

### Official Review · Reviewer_HVjD · 2025-11-01

**Soundness:** 3
**Presentation:** 3
**Contribution:** 3
**Rating:** 4
**Confidence:** 4

**Summary:**

This paper proposes MIDAS, a multimodal human sensing framework designed to robustly handle missing modalities by addressing two fundamental challenges: the Representation Gap between heterogeneous sensors (e.g., depth vs. WiFi CSI) and the Contamination Effect, where low-quality modalities degrade fusion performance. To overcome these issues, MIDAS introduces two key components: (1) a meta-learning-based modality weighting mechanism that adaptively learns optimal fusion weights through bi-level optimization over training and validation data, and (2) a diffusion-based knowledge distillation strategy, where features fused from all available modalities form a “consensus teacher” that guides and refines each single-modality feature via diffusion denoising. Experiments on the MM-Fi dataset for human pose estimation and XRF55 dataset for action recognition demonstrate state-of-the-art performance across a wide range of missing-modality scenarios, and ablation studies confirm the effectiveness of both the diffusion-based feature alignment and meta-learning weighting modules.

**Strengths:**

1. Clear and well-organized presentation
The paper is clearly written and well-structured. It provides a coherent problem definition by identifying the Representation Gap and Contamination Effect as two core challenges in multimodal human sensing with missing modalities. The methodology is introduced progressively and logically, with clear mathematical formulations, diagrams, and detailed explanations of the meta-learning optimization and diffusion-based distillation processes. This clarity significantly improves the readability and reproducibility of the work.

2. Comprehensive and rigorous ablation studies
The authors conduct thorough ablation experiments to validate the effectiveness of each component in the proposed framework. By separately removing the diffusion-based alignment and the meta-learning weighting module, the paper clearly demonstrates the contribution of each part to overall performance. The experiments cover single-modality, partial-modality, and full-modality settings on two datasets, which shows that the authors have carefully evaluated how each module behaves under different conditions. This enhances the credibility and robustness of the proposed approach.

**Weaknesses:**

1. Insufficient comparison with recent state-of-the-art methods
This paper does not include comparisons with more recent and competitive approaches in multimodal learning, knowledge distillation, or diffusion-based alignment. The absence of these comparisons makes it difficult to fully position the proposed method within the current research landscape.

2. Relationship between the two core modules is not well justified
MIDAS integrates two major components: meta-learning-based modality weighting and diffusion-based knowledge distillation. However, the paper does not provide a strong explanation of why these two modules must be combined in a single framework, or how they conceptually and theoretically interact. It is not entirely clear whether these two mechanisms are interdependent or simply operate in parallel. This gives the framework a somewhat modular or “stitched-together” feeling rather than a tightly unified design. A deeper justification for their integration would strengthen the paper.

3. Limited discussion on generalization and applicability to broader domains
All experiments are conducted within the domain of human sensing using sensor data such as depth, LiDAR, WiFi, mmWave, and RFID. While results are strong in this area, it remains unclear whether the proposed method generalizes to other multimodal tasks, such as vision-language learning, audio-visual reasoning, or multimodal medical imaging. The datasets used are relatively homogeneous in task type and sensor modality. Without experiments or discussion on cross-domain applicability, the generalization ability and universality of the proposed method remains uncertain.

**Questions:**

please see weaknesses

---

### Official Review · Reviewer_K4zt · 2025-11-06

**Soundness:** 2
**Presentation:** 2
**Contribution:** 2
**Rating:** 4
**Confidence:** 3

**Summary:**

This paper proposes MIDAS, a novel framework for robust multimodal human sensing under missing modality conditions. MIDAS addresses two core challenges: the Representation Gap between heterogeneous sensor data and the Contamination Effect caused by low-quality modalities. To bridge the representation gap, it introduces a diffusion-based knowledge distillation mechanism where a consensus teacher (fused from all available modalities) guides the refinement of each individual "student" modality. To mitigate contamination, it employs a meta-learning-based weighting mechanism that dynamically assigns importance weights to modalities based on their contribution to task performance. Evaluated on MM-Fi and XRF55 datasets for human pose estimation and action recognition, MIDAS demonstrates state-of-the-art performance across various missing-modality scenarios.

**Strengths:**

- Synergistic Dual Mechanism: The integration of meta-learning for dynamic modality weighting and diffusion-based feature alignment provides a principled solution to both the Contamination Effect and Representation Gap simultaneously.

- Effective Feature Refinement: The proposed "many-to-one" diffusion-based distillation effectively enhances the quality of single-modality features by aligning them with a richer, fused teacher representation, significantly boosting standalone performance.

- Parameter Efficiency and Scalability: Unlike existing methods whose complexity scales with the number of modalities, MIDAS maintains a fixed, compact parameter count, demonstrating superior scalability and efficiency.

**Weaknesses:**

- Potential for Degradation in Specific Fusion Scenarios: The ablation study reveals that removing the diffusion module can paradoxically improve performance in certain combinations involving very low-quality modalities, suggesting the generative process may introduce noise or artifacts.

- Inadequate Analysis of Meta-Weighting Behavior: The paper lacks a detailed investigation into how the learned weights evolve during training and how they respond to varying degrees of modality corruption or synthetic noise injection.

- Limited Generalizability Beyond RF-Dominated Modalities: The significant gains on XRF55 (all RF-based) compared to MM-Fi suggest the method's effectiveness might be contingent on a relatively smaller inherent representation gap between modalities.

- Computational Overhead During Training: The use of nested bi-level optimization and a diffusion model introduces substantial computational complexity, which is only briefly mentioned as being deferred to supplementary material.

**Questions:**

- Comment 1: The observation that performance improves without the diffusion module in some cases (e.g., W+RF on XRF55) indicates a fundamental risk: the reverse diffusion process, while intended for purification, can inadvertently amplify noise or generate misleading features when conditioned on highly degraded student inputs, potentially undermining its purpose.

- Comment 2: The paper fails to provide an analysis of the final learned modality weights. Without visualizing these weights or correlating them with known modality signal-to-noise ratios, it remains unclear whether the meta-learning process has converged to a meaningful and interpretable policy or an opaque heuristic.

- Comment 3: While the framework claims to mitigate contamination, the paper does not explore the limits of this capability. It leaves unanswered the critical question of what level of modality degradation renders fusion detrimental, despite acknowledging this as a future direction.

- Comment 4: The experiments do not demonstrate the system's ability to adapt weights dynamically to changing data quality. A more rigorous test would involve introducing time-varying noise to a modality and showing the weights adjust accordingly, which is absent from the evaluation.

- Comment 5: The superior results on the RF-centric XRF55 dataset, contrasted with more modest gains on the diverse-sensor MM-Fi dataset, raises concerns about the method's general applicability to systems combining radically different sensing principles (e.g., vision, audio, inertial).

- Comment 6: The requirement for computationally expensive bi-level optimization and diffusion model training creates a significant practical barrier for replication and deployment, especially given that the reported inference overhead is minimal. This trade-off is not thoroughly justified.

- Comment 7: The work presents a powerful empirical framework but offers limited theoretical insight into why the combination of meta-learning and diffusion is effective for this specific problem, leaving the underlying mechanics somewhat heuristic.

**Details Of Ethics Concerns:**

None.

---

### Note · Authors · 2025-11-15

**Comment:**

Thank you for the valuable feedback. We appreciate your time and effort. We have decided to revise the paper based on the comments received and plan to submit an improved version to a future venue.

**Withdrawal Confirmation:**

I have read and agree with the venue's withdrawal policy on behalf of myself and my co-authors.